# Adversarial Attacks on Stochastic Bandits

**Kwang-Sung Jun**
Boston University
kwangsung.jun@gmail.com

**Lihong Li**
Google Brain
lihong@google.com

**Yuzhe Ma**
UW-Madison
ma234@wisc.edu

**Xiaojin Zhu**
UW-Madison
jerryzhu@cs.wisc.edu

## Abstract

We study adversarial attacks that manipulate the reward signals to control the actions chosen by a stochastic multi-armed bandit algorithm. We propose the first attack against two popular bandit algorithms: $\epsilon$-greedy and UCB, *without* knowledge of the mean rewards. The attacker is able to spend only logarithmic effort, multiplied by a problem-specific parameter that becomes smaller as the bandit problem gets easier to attack. The result means the attacker can easily hijack the behavior of the bandit algorithm to promote or obstruct certain actions, say, a particular medical treatment. As bandits are seeing increasingly wide use in practice, our study exposes a significant security threat.

## 1   Introduction

Designing trustworthy machine learning systems requires understanding how they may be attacked. There has been a surge of interest on adversarial attacks against supervised learning [12, 15]. In contrast, little is known on adversarial attacks against stochastic multi-armed bandits (MABs), a form of online learning with limited feedback. This is potentially hazardous since stochastic MABs are widely used in the industry to recommend news articles [18], display advertisements [9], improve search results [17], allocate medical treatment [16], and promote users' well-being [13], among many others. Indeed, as we show, an adversarial attacker can modify the reward signal to manipulate the MAB for nefarious goals.

Our main contribution is an analysis on reward-manipulation attacks. We distinguish three agents in this setting: "the world," "Bob" the bandit algorithm, and "Alice" the attacker. As in standard stochastic bandit problems, the world consists of $K$ arms with sub-Gaussian rewards centered at $\mu_1, \ldots, \mu_K$. Note that we do *not* assume $\{\mu_i\}$ are sorted. Neither Bob nor Alice knows $\{\mu_i\}$. Bob pulls selected arms in rounds and attempts to minimize his regret. When Bob pulls arm $I_t \in [K]$ in round $t$, the world generates a random reward $r_t^0$ drawn from a sub-Gaussian distribution with expectation $\mu_{I_t}$. However, Alice sits in-between the world and Bob and manipulates the reward into $r_t = r_t^0 - \alpha_t$. We call $\alpha_t \in \mathbb{R}$ the attack. If Alice decides not to attack in this round, she simply lets $\alpha_t = 0$. Bob then receives $r_t$, without knowing the presence of Alice. Without loss of generality, assume arm $K$ is a suboptimal "attack target" arm: $\mu_K < \max_{i=1...K} \mu_i$. Alice's goal is to manipulate Bob into pulling arm $K$ very often while making small attacks. Specifically, we show Alice can force Bob to pull the target arm $T - o(T)$ number of times with a cumulative attack cost of $\sum_{t=1}^{T} |\alpha_t| = O(\log(T))$.

The assumption that Alice does not know $\{\mu_i\}$ is significant because otherwise Alice can perform the attack trivially. To see this, with the knowledge of $\{\mu_i\}$ Alice would be able to compute the truncated reward gap $\Delta_i^\epsilon = \max\{\mu_i - \mu_K + \epsilon, 0\} \geq 0$ for all non-target arms $i \neq K$ for some small parameter

$\epsilon > 0$. Alice can perform the following *oracle attack*: in any round where a non-target arm $I_t \neq K$ is pulled, attack with $\alpha_t = \Delta_{I_t}^\epsilon$. This oracle attack transforms the original bandit problem into one where all non-target arms have expected reward less than $\mu_K$. It is well-known that if Bob runs a sublinear-regret algorithm (e.g., UCB [6, 8]), almost all arm pulls will concentrate on the now-best target arm $K$ in the transformed bandit problem. Furthermore, Alice's cumulative attack cost will be sublinear in time, because the total number of non-target arm pulls is sublinear in the transformed problem. In practice, however, it is almost never the case that Alice knows $\mu_1, \ldots, \mu_K$ and hence the $\Delta_i^\epsilon$'s. Thus the oracle attack is impractical. Our focus in this paper is to design an attack that nearly matches the oracle attack, but for Alice who does not know $\{\mu_i\}$. We do so for two popular bandit algorithms, $\epsilon$-greedy [7] and UCB [8].

What damage can Alice do in practice? She can largely control the arms pulled by Bob. She can also control which arm appears to Bob as the best arm at the end. As an example, consider the news-delivering contextual bandit problem [18]. The arms are available news articles, and Bob selects which arm to pull (i.e., which article to show to a user at the news site). In normal operation, Bob shows news articles to users to maximize the click-through rate. However, Alice can attack Bob to change his behavior. For instance, Alice can manipulate the rewards so that users from a particular political base are always shown particular news articles that can reinforce or convert their opinion. Conversely, Alice can coerce the bandit to not show an important article to certain users. As another example, Alice may interfere with clinical trials [16] to funnel most patients toward certain treatment, or make researchers draw wrong conclusions on whether treatment is better than control. Therefore, adversarial attacks on MAB deserve our attention. Insights gained from our study can be used to build defense in the future.

Finally, we note that our setting is motivated by modern industry-scale applications of contextual bandits, where arm selection, reward signal collection, and policy updates are done in a distributed way [3, 18]. Attacks can happen when the reward signal is joined with the selected arm, or when the arm-reward data is sent to another module for Bob to update his policy. In either case, Alice has access to both $I_t$ and $r_t^0$ for the present and previous rounds.

The rest of the paper is organized as follows. In Section 2, we introduce notations and straightforward attack algorithms that serve as baseline. We then propose our two attack algorithms for $\epsilon$-greedy and UCB in Section 3 and 4 respectively, along with their theoretical attack guarantees. In Section 5, we empirically confirm our findings with toy experiments. Finally, we conclude our paper with related work (Section 6) and a discussion of future work (Section 7) that will enrich our understanding of security vulnerability and defense mechanisms for secure MAB deployment.

## 2  Preliminaries

Before presenting our main attack algorithms, in this section we first discuss a simple heuristic attack algorithm which serves to illustrate the intrinsic difficulty of attacks. Throughout, we assume Bob runs a bandit algorithm with sublinear pseudo-regret $\mathbb{E} \sum_{t=1}^T (\max_{j=1}^K \mu_j - \mu_{I_t})$. As Alice does not know $\{\mu_i\}$ she must rely on the empirical rewards up to round $t - 1$ to decide the appropriate attack $\alpha_t$. The attack is online since $\alpha_t$ is computed on-the-fly as $I_t$ and $r_t^0$ are revealed. The attacking protocol is summarized in Algorithm 1.

---
**Algorithm 1** Alice's attack against a bandit algorithm
---
1: **Input**: Bob's bandit algorithm, target arm $K$
2: **for** $t = 1, 2, \ldots$ **do**
3:     Bob chooses arm $I_t$ to pull.
4:     World generates pre-attack reward $r_t^0$.
5:     Alice observes $I_t$ and $r_t^0$, and then decides the attack $\alpha_t$.
6:     Alice gives $r_t = r_t^0 - \alpha_t$ to Bob.
7: **end for**
---

We assume all arm rewards are $\sigma^2$-sub-Gaussian where $\sigma^2$ is known to both Alice and Bob. Let $N_i(t)$ be the number of pulls of arm $i$ up to round $t$. We say the attack is *successful* after $T$ rounds if the

number of target-arm pulls is $N_K(T) = T - o(T)$ in expectation or with high probability, while minimizing the cumulative attack cost $\sum_{t=1}^{T} |\alpha_t|$. Other attack settings are discussed in Section 7.

For convenience we define the following quantities:

- $\tau_i(t) := \{s : s \le t, I_s = i\}$, the set of rounds up to $t$ where arm $i$ is chosen,
- $\hat{\mu}_i^0(t) := N_i(t)^{-1} \sum_{s \in \tau_i(t)} r_s^0$, the pre-attack average reward of arm $i$ up to round $t$, and
- $\hat{\mu}_i(t) := N_i(t)^{-1} \sum_{s \in \tau_i(t)} r_s$, the corresponding post-attack average reward.

**The oracle attack, revisited**    While the oracle attack was impractical, it gives us a baseline for comparison. The oracle attack drags down the reward of all non-target arms,[1] and can be written as

$$\alpha_t = \mathbb{1}\{I_t \ne K\} \cdot \Delta_{I_t}^\epsilon .$$

Proposition 1 shows that the oracle attack succeeds and requires only a logarithmic attack cost. While more general statements for sublinear-regret algorithms can be made, we focus on logarithmic-regret bandit algorithms for simplicity. Throughout, omitted proofs can be found in our supplementary material.

**Proposition 1.** *Assume that Bob's bandit algorithm achieves an $O(\log T)$ regret bound. Then, Alice's oracle attack with $\epsilon > 0$ succeeds; i.e., $\mathbb{E} N_K(T) = T - o(T)$. Furthermore, the expected total attack cost is $O\left(\sum_{i=1}^{K-1} \Delta_i^\epsilon \log T\right)$.*[2]

**The heuristic constant attack**    A slight variant of the oracle attack is to attack all the non-target arms with a single constant amount $A > 0$, regardless of the actual $\mu_i$'s:

$$\alpha_t = \mathbb{1}\{I_t \ne K\} \cdot A.$$

Let $\Delta_i := \Delta_i^0$. Unfortunately, this heuristic constant attack depends critically on the value of $A$ compared to the unknown maximum gap $\max_i \Delta_i$. Proposition 2 states the condition under which the attack succeeds:

**Proposition 2.** *Assume that Bob's bandit algorithm achieves an $O(\log T)$ regret bound. Then, Alice's heuristic constant attack with $A$ succeeds if and only if $A > \max_i \Delta_i$. If the attack succeeds, then the expected attack cost is $O(AK \log T)$.*

Conversely, if $A < \max_i \Delta_i$ the attack fails. This is because in the transformed bandit problem, there exists an arm that has a higher expected reward than arm $K$, and Bob will mostly pull that arm. Therefore, the heuristic constant attack has to know an unknown quantity to guarantee a successful attack. Moreover, the attack is non-adaptive to the problem difficulty since some $\Delta_i$'s can be much smaller than $A$, in which case Alice pays an unnecessarily large attack cost.

We therefore ask the following question:

> Does there exist an attacker Alice that guarantees a successful attack with cost adaptive to the problem difficulty?

The answer is yes. We present attack strategies against two popular bandit algorithms of Bob: $\epsilon$-greedy and UCB. We show that Alice can indeed succeed in her attacks and incur cost as small as that of the oracle with an additive term due to the sub-Gaussian noise level $\sigma$.

## 3    Alice's Attack on $\epsilon$-Greedy Bob

The $\epsilon$-greedy strategy initially pulls each arm once in the first $K$ rounds. For convenience, we assume that the target arm is pulled first: $I_1 = K$. Our results in this section can be adapted to any order of initialization with more complicated notation.

Bob's $\epsilon$-greedy strategy has the following arm-selection rule for $t > K$ [7]:

$$I_t = \begin{cases} \text{draw uniform}[K], & \text{w.p. } \epsilon_t \quad \text{(exploration)} \\ \arg\max_i \hat{\mu}_i(t-1), & \text{otherwise (exploitation)} \end{cases}.$$

The strategy uses an exploration scheme $\{\epsilon_t\}$ over $t$. Alice's attack algorithm is not aware of $\{\epsilon_t\}$ though her cumulative attack cost $\sum |\alpha_t|$ will implicitly depend on it. Later in Corollary 1 we show that, for the typical decaying scheme $\epsilon_t \propto 1/t$, the cumulative attack cost is mild: $O(\log(t))$.

Alice wants to make Bob *always* pull the target arm during *exploitation* rounds. Since Alice has no influence on which arm is pulled during exploration, this attack goal is the strongest she can achieve. Here, Algorithm 1 is specialized to ensure the following condition:

$$\hat{\mu}_{I_t}(t) \leq \hat{\mu}_K(t) - 2\beta(N_K(t)), \tag{1}$$

where we define $\beta(N)$ as

$$\beta(N) := \sqrt{\frac{2\sigma^2}{N} \log \frac{\pi^2 K N^2}{3\delta}}. \tag{2}$$

From this condition, we derive the actual attack $\alpha_t$. Since

$$\hat{\mu}_{I_t}(t) = \frac{\hat{\mu}_{I_t}(t-1)N_{I_t}(t-1) + r_t^0 - \alpha_t}{N_{I_t}(t)}, \tag{3}$$

we set the attack in Algorithm 1 as

$$\alpha_t = \left[ \hat{\mu}_{I_t}(t-1)N_{I_t}(t-1) + r_t^0 - (\hat{\mu}_K(t) - 2\beta(N_K(t))) N_{I_t}(t) \right]_+, \tag{4}$$

where $[z]_+ = \max(0, z)$. Note $\alpha$ is always non-negative, thus the cumulative attack cost can be written without absolute value: $\sum_{t=1}^T \alpha_t$.

With this $\alpha_t$, we claim that (i) Alice forces Bob to pull the target arm in all exploitation rounds as shown in Lemma 2, and (ii) the cumulative attack cost is logarithmic in $t$ for standard $\epsilon$-greedy learner exploration scheme $\epsilon_t = O(1/t)$ as shown in Corollary 1. Our main result is the following general upper bound on the cumulative attack cost.

**Theorem 1.** *Let $\delta \leq 1/2$. With probability at least $1 - 2\delta$, for any $T$ satisfying $\sum_{t=1}^T \epsilon_t \geq \frac{K}{e-2} \log(K/\delta)$,[3] Alice forces Bob running $\epsilon$-greedy to choose the target arm in at least $\widetilde{N}_K(T)$ rounds, using a cumulative attack cost at most*

$$\sum_{t=1}^T |\alpha_t| < \left( \sum_{i=1}^K \Delta_i \right) \widetilde{N}(T) + (K-1) \cdot \left( \widetilde{N}(T)\beta(\widetilde{N}(T)) + 3\widetilde{N}(T)\beta(\widetilde{N}_K(T)) \right)$$

*where*

$$\widetilde{N}(T) = \left( \frac{\sum_{t=1}^T \epsilon_t}{K} \right) + \sqrt{3\log\left(\frac{K}{\delta}\right)\left(\frac{\sum_{t=1}^T \epsilon_t}{K}\right)},$$

$$\widetilde{N}_K(T) = T - \left( \sum_{t=1}^T \epsilon_t \right) - \sqrt{3\log\left(\frac{K}{\delta}\right)\left(\sum_{t=1}^T \epsilon_t\right)}.$$

Before proving the theorem, we first look at its consequence. If Bob's $\epsilon_t$ decay scheme is $\epsilon_t = \min\{1, cK/t\}$ for some $c > 0$ as recommended in Auer et al. [7], Alice's cumulative attack cost is $O(\sum_{i=1}^K \Delta_i \log T)$ for large enough $T$, as the following corollary shows:

**Corollary 1.** *Inherit the assumptions in Theorem 1. Fix $K$ and $\delta$. If $\epsilon_t = cK/t$ for some constant $c > 0$, then*

$$\sum_{t=1}^T |\alpha_t| = \widehat{O}\left( \left( \sum_{i=1}^K \Delta_i \right) \log T + \sigma K \sqrt{\log T} \right), \tag{5}$$

*where $\widehat{O}$ ignores* log log *factors.*

Note that the two important constants are $\sum_i \Delta_i$ and $\sigma$. While a large $\sigma$ can increase the cost significantly, the term with $\sum_i \Delta_i$ dominates the cost for large enough $T$. Specifically, $\sum_i \Delta_i$ is multiplied by $\log T$ that is of higher order than $\sqrt{\log T}$. We empirically verify the scaling of cost with $T$ in Section 5.

To prove Theorem 1, we first show that $\beta$ in (2) is a high-probability bound on the pre-attack empirical mean of all arms on all rounds. Define the event

$$E := \{\forall i, \forall t > K : |\hat{\mu}_i^0(t) - \mu_i| < \beta(N_i(t))\}. \tag{6}$$

**Lemma 1.** *For $\delta \in (0,1)$, $\mathbb{P}(E) > 1 - \delta$.*

The following lemma proves the first half of our claim.

**Lemma 2.** *For $\delta \le 1/2$ and under event $E$, attacks (4) force Bob to always pull the target arm $K$ in exploitation rounds.*

We now show that on average each attack on a non-target arm $i$ is not much bigger than $\Delta_i$.

**Lemma 3.** *For $\delta \le 1/2$ and under event $E$, we have for all arm $i < K$ and all $t$ that*

$$\sum_{s \in \tau_i(t)} |\alpha_s| < (\Delta_i + \beta(N_i(t)) + 3\beta(N_K(t))) \, N_i(t).$$

Finally, we upper bound the number of non-target arm $i$ pulls $N_i(T)$ for $i < K$. Recall the arm $i$ pulls are only the result of exploration rounds. In round $t$ the exploration probability is $\epsilon_t$; if Bob explores, he chooses an arm uniformly at random. We also lower bound the target arm pulls $N_K(T)$.

**Lemma 4.** *Let $\delta < 1/2$. Suppose $T$ satisfy $\sum_{t=1}^{T} \epsilon_t \ge \frac{K}{e-2} \log(K/\delta)$. With probability at least $1 - \delta$, for all non-target arms $i < K$,*

$$N_i(T) < \sum_{t=1}^{T} \frac{\epsilon_t}{K} + \sqrt{3 \sum_{s=1}^{T} \frac{\epsilon_t}{K} \log \frac{K}{\delta}}.$$

*and for the target arm $K$,*

$$N_K(T) > T - \sum_{t=1}^{T} \epsilon_t - \sqrt{3 \sum_{s=1}^{T} \epsilon_t \log \frac{K}{\delta}}.$$

We are now ready to prove Theorem 1.

*Proof.* The theorem follows immediately from a union bound over Lemma 3 and Lemma 4 below. We add up the attack costs over $K - 1$ non-target arms. Then, we note that $N\beta(N)$ is increasing in $N$ so $N_i(T)\beta(N_i(T)) \le \widetilde{N}(T)\beta(\widetilde{N}(T))$. Finally, by Lemma 8 in our supplementary material $\beta(N)$ is decreasing in $N$, so $\beta(N_K(T)) \le \beta(\widetilde{N}_K(T))$. $\qquad \square$

## 4 Alice's Attack on UCB Bob

Recall that we assume rewards are $\sigma^2$-sub-Gaussian. Bob's UCB algorithm in its basic form often assumes rewards are bounded in $[0, 1]$; we need to modify the algorithm to handle the more general sub-Gaussian rewards. By choosing $\alpha = 4.5$ and $\psi : \lambda \mapsto \frac{\sigma^2 \lambda^2}{2}$ in the $(\alpha, \psi)$-UCB algorithm of Bubeck & Cesa-Bianchi [8, Section 2.2], we obtain the following arm-selection rule:

$$I_t = \begin{cases} t, & \text{if } t \le K \\ \arg\max_i \left\{ \hat{\mu}_i(t-1) + 3\sigma\sqrt{\frac{\log t}{N_i(t-1)}} \right\}, & \text{otherwise.} \end{cases}$$

For the first $K$ rounds where Bob plays each of the $K$ arms once in an arbitrary order, Alice does not attack: $\alpha_t = 0$ for $t \le K$. After that, attack happens only when $I_t \ne K$. Specifically, consider any round $t > K$ where Bob pulls arm $i \ne K$. It follows from the UCB algorithm that

$$\hat{\mu}_i(t-1) + 3\sigma\sqrt{\frac{\log t}{N_i(t-1)}} \ge \hat{\mu}_K(t-1) + 3\sigma\sqrt{\frac{\log t}{N_K(t-1)}}.$$

Alice attacks as follows. She computes an attack $\alpha_t$ with the smallest absolute value, such that

$$\hat{\mu}_i(t) \leq \hat{\mu}_K(t-1) - 2\beta(N_K(t-1)) - \Delta_0,$$

where $\Delta_0 \geq 0$ is a parameter of Alice. Since the post-attack empirical mean can be computed recursively by the following

$$\hat{\mu}_i(t) = \frac{N_i(t-1)\hat{\mu}_i(t-1) + r_t^0 - \alpha_t}{N_i(t-1) + 1},$$

where $r_t^0$ is the pre-attack reward; this enables us to write down in closed form Alice's attack:

$$\alpha_t = \left[ N_i(t)\hat{\mu}_i^0(t) - \sum_{s \in \tau_i(t-1)} \alpha_s - N_i(t) \cdot (\hat{\mu}_K(t-1) - 2\beta(N_K(t-1)) - \Delta_0) \right]_+. \quad (7)$$

For convenience, define $\alpha_t = 0$ if $I_t = K$. We now present the main theorem on Alice's cumulative attack cost against Bob who runs UCB.

**Theorem 2.** *Suppose $T \geq 2K$ and $\delta \leq 1/2$. Then, with probability at least $1 - \delta$, Alice forces Bob to choose the target arm in at least*

$$T - (K-1)\left(2 + \frac{9\sigma^2}{\Delta_0^2}\log T\right),$$

*rounds, using a cumulative attack cost at most*

$$\sum_{t=1}^{T} \alpha_t \leq \left(2 + \frac{9\sigma^2}{\Delta_0^2}\log T\right)\sum_{i<K}(\Delta_i + \Delta_0) + \sigma(K-1)\sqrt{32\left(2 + \frac{9\sigma^2}{\Delta_0^2}\log T\right)\log\frac{\pi^2 K(2 + \frac{9\sigma^2}{\Delta_0^2}\log T)^2}{3\delta}}.$$

While the bounds in the theorem are somewhat complicated, the next corollary is more interpretable and follows from a straightforward calculation.

**Corollary 2.** *Inherit the assumptions in Theorem 2 and fix $\delta$. Then, the total number of non-target arm pulls is*

$$O\left(K + \frac{K\sigma^2}{\Delta_0^2}\log T\right),$$

*and the cumulative attack cost is*

$$\widehat{O}\left(\left(1 + \frac{\sigma^2}{\Delta_0^2}\log T\right)\sum_{i<K}(\Delta_i + \Delta_0) + \sigma K \cdot \left(1 + \frac{\sigma}{\Delta_0}\sqrt{\log T}\right)\sqrt{\log\left(1 + \frac{K\sigma}{\Delta_0}\right)}\right),$$

*where $\widehat{O}$ ignores $\log\log(T)$ factors.*

We observe that a larger $\Delta_0$ decreases non-target arm pulls (i.e. a more effective attack). The effect diminishes when $\Delta_0 > \sigma\sqrt{\log T}$ since $\frac{K\sigma^2}{\Delta_0^2}\log T < K$. Thus there is no need for Alice to choose a larger $\Delta_0$. By choosing $\Delta_0 = \Theta(\sigma)$, the cost is $\widehat{O}(\sum_{i<K}\Delta_i \log T + \sigma K \log T)$. This is slightly worse than the cost of attacking $\epsilon$-greedy where $\sigma$ is multiplied by $\sqrt{\log T}$ rather than $\log T$. However, we find that a stronger attack is possible when the time horizon $T$ is fixed and known to Alice ahead of time (i.e., the fixed budget setting). One can show that the choice $\Delta_0 = \Theta(\sigma\sqrt{\log T})$ minimizes the cumulative attack cost, which is $\widehat{O}(K\sigma\sqrt{\log T})$. This is a very efficient attack since the dominating term w.r.t. $T$ does not depend on $\sum_{i<K}\Delta_i$; in fact the cost associated with $\sum_{i<K}\Delta_i$ does not grow with $T$ at all. Furthermore, such a cost matches the cost of the oracle attack up to doubly-logarithmic factors, which is shown in our supplementary material A.

For the proof of Theorem 2 we use the following two lemmas.

**Lemma 5.** *Assume event $E$ holds and $\delta \leq 1/2$. Then, for any $i < K$ and any $t \geq 2K$, we have*

$$N_i(t) \leq \min\{N_K(t), 2 + \frac{9\sigma^2}{\Delta_0^2}\log t\}. \quad (8)$$

**Lemma 6.** *Assume event $E$ holds and $\delta \leq 1/2$. Then, at any round $t \geq 2K$, the cumulative attack cost to any fixed arm $i < K$ can be bounded as:*

$$\sum_{s \in \tau_i(t)} \alpha_s \leq N_i(t)\left(\Delta_i + \Delta_0 + 4\beta(N_i(t))\right).$$

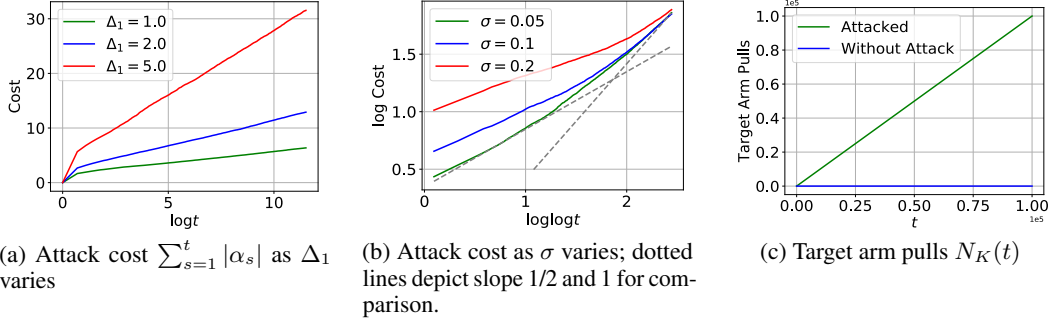

(a) Attack cost $\sum_{s=1}^{t} |\alpha_s|$ as $\Delta_1$ varies

(b) Attack cost as $\sigma$ varies; dotted lines depict slope 1/2 and 1 for comparison.

(c) Target arm pulls $N_K(t)$

Figure 1: Attack on $\epsilon$-greedy bandit.

*Proof of Theorem* 2. Suppose event $E$ holds. The bounds are direct consequences of Lemmas 6 and 5 below, by summing the corresponding upper bounds over all non-target arms $i$. Specifically, the number of target arm pulls is $T - \sum_{i<K} N_i(T)$, and the cumulative attack cost is $\sum_{t=1}^{T} \alpha_t = \sum_{i<K} \sum_{t \in \tau_i(T)} \alpha_t$. Since event $E$ is true with probability at least $1 - \delta$ (Lemma 1), the bounds also hold with probability at least $1 - \delta$. □

## 5 Simulations

In this section, we run simulations on attacking $\epsilon$-greedy and UCB algorithms to illustrate our theoretical findings.

**Attacking $\epsilon$-greedy** The bandit has two arms. The reward distributions of arms 1 and 2 are $\mathcal{N}(\Delta_1, \sigma^2)$ and $\mathcal{N}(0, \sigma^2)$, respectively, with $\Delta_1 > 0$. Alice's target arm is arm 2. We let $\delta = 0.025$. Bob's exploration probability decays as $\epsilon_t = \frac{1}{t}$. We run Alice and Bob for $T = 10^5$ rounds; this forms one trial. We repeat 1000 trials.

In Figure 1(a), we fix $\sigma = 0.1$ and show Alice's cumulative attack cost $\sum_{s=1}^{t} |\alpha_s|$ for different $\Delta_1$ values. Each curve is the average over 1000 trials. These curves demonstrate that Alice's attack cost is proportional to $\log t$ as predicted by Corollary 1. As the reward gap $\Delta_1$ becomes larger, more attack is needed to reduce the reward of arm 1, and the slope increases.

Furthermore, note that $\sum_{t=1}^{T} |\alpha_t| = \widehat{O}\left(\Delta_1 \log T + \sigma\sqrt{\log T}\right)$. Ignoring $\log\log T$ terms, we have $\sum_{t=1}^{T} |\alpha_t| \leq C(\Delta_1 \log T + \sigma\sqrt{\log T})$ for some constant $C > 0$ and large enough $T$. Therefore, $\log\left(\sum_{t=1}^{T} |\alpha_t|\right) \leq \max\{\log\log T + \log \Delta_1, \frac{1}{2}\log\log T + \log \sigma\} + \log C$. We thus expect the log-cost curve as a function of $\log\log T$ to behave like the maximum of two lines, one with slope 1/2 and the other with slope 1. Indeed, we observe such a curve in Figure 1(b) where we fix $\Delta_1 = 1$ and vary $\sigma$. All the slopes eventually approach 1, though larger $\sigma$'s take a longer time. This implies that the effect of $\sigma$ diminishes for large enough $T$, which was predicted by Corollary 1.

In Figure 1(c), we compare the number of target arm (the suboptimal arm 2) pulls with and without attack. This experiment is with $\Delta_1 = 0.1$ and $\sigma = 0.1$. Alice's attack dramatically forces Bob to pull the target arm. In 10000 rounds, Bob is forced to pull the target arm 9994 rounds with the attack, compared to only 6 rounds if Alice was not present.

**Attacking UCB** The bandit has two arms. The reward distributions are the same as the $\epsilon$-greedy experiment. We let $\delta = 0.05$. To study how $\sigma$ and $\Delta_0$ affects the cumulative attack cost, we perform two groups of experiments. In the first group, we fix $\sigma = 0.1$ and vary Alice's free parameter $\Delta_0$ while in the second group, we fix $\Delta_0 = 0.1$ and vary $\sigma$. We perform 100 trials with $T = 10^7$ rounds.

Figure 2(a) shows Alice's cumulative attack cost as $\Delta_0$ varies. As $\Delta_0$ increases, the cumulative attack cost decreases. In Figure 2(b), we show the cost as $\sigma$ varies. Note that for large enough $t$, the cost grows almost linearly with $\log t$, which is implied by Corollary 2. In both figures, there is a large attack near the beginning, after which the cost grows slowly. This is because the initial attacks drag down the empirical average of non-target arms by a large amount, such that the target arm appears to have the best UCB for many subsequent rounds. Figure 2(c) again shows that Alice's attack forces

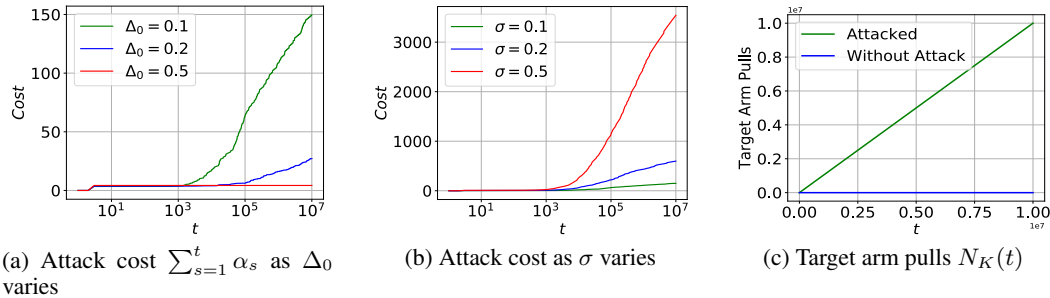

(a) Attack cost $\sum_{s=1}^{t} \alpha_s$ as $\Delta_0$ varies

(b) Attack cost as $\sigma$ varies

(c) Target arm pulls $N_K(t)$

Figure 2: Attack on UCB learner.

Bob to pull the target arm: with attack Bob is forced to pull the target arm $10^7 - 2$ times, compared to only $156$ times without attack.

## 6 Related Work

The literature on general adversarial learning is vast and covers ethics, safety, fairness, and legal concerns; see e.g. Joseph et al. [15] and Goodfellow et al. [12]. Related to MAB, there has been empirical evidence that suggests adversarial attacks can be quite effective, even in the more general multi-step reinforcement learning problems, as opposed to the bandit case considered in this paper. The learned policy may be lured to visit certain target states when adversarial examples are driven [19], or have inferior generalization ability when training examples are corrupted [14]. There are differences, though. In the first, non-stochastic setting [7, 11], the reward is generated by an adversary instead of a stationary, stochastic process. However, the reward observed by the learner is still a *real* reward, in that the learner is still interested in maximizing it, or more precisely, minimizing some notion of regret in reference to some reference policy [8]. Another related problem is reward shaping (e.g., Dorigo & Colombetti [10]), where the reward received by the learner is modified, as in our paper. However, those changes are typically done to *help* the learner in various ways (such as promoting exploration), and are designed in a way not to change the optimal policy the learner eventually converges to [22].

A concurrent work by Lykouris et al. [20] considers a complementary problem to ours. They propose a randomized bandit algorithm that is robust to adversarial attacks on the stochastic rewards. In contrast, our work shows that the existing stochastic algorithms are vulnerable to adversarial attacks. Note that their attack protocol is slightly different in that the attacker has to prepare attacks for all the arms before the learner chooses an arm. Furthermore, they have a different attack cost definition where the cost in a round is the *largest* manipulation over the arms, regardless of which arm the learner selects afterwards.

Another concurrent work by Ma et al. [21] considers attacking stochastic contextual bandit algorithms. The authors show that for a contextual bandit algorithm which periodically updates the arm selection policy, an attacker can perform offline attack to force the contextual bandit algorithm to pull some pre-specified target arm for a given target context vector. Our work differs in that we consider online attack, which is performed on the fly rather than offline.

## 7 Conclusions and Future Work

We presented a reward-manipulating attack on stochastic MABs. We analyzed the attack against $\epsilon$-greedy and a generalization of the UCB algorithm, and proved that the attacker can force the algorithms to almost always pull a suboptimal target arm. The cost of the attack is only logarithmic in time. Given the wide use of MABs in practice, this is a significant security threat.

Our analysis is only the beginning. We targeted $\epsilon$-greedy and UCB learners for their simplicity and popularity. Future work may look into attacking Thompson sampling [23, 4], linear bandits [1, 5], and contextual bandits [18, 2], etc. We assumed the reward attacks $\alpha_t$ are unbounded from above; new analysis is needed if an application's reward space is bounded or discrete. It will also be useful

to establish lower bounds on the cumulative attack cost. Specifically, it would be interesting to study pareto optimality w.r.t. the number of target arm pulls and the cumulative attack cost.

Beyond the attack studied in this paper, there is a wide range of possible attacks on MABs. We may organize them along several dimensions:

- Optimal control viewpoint: Our 'reward shaping' attack model can be formulated as optimal control [24]. We can define the control cost as $\alpha_t^2 + \lambda \mathbb{1}\{I_t \neq K\}$ and design optimal control strategies.
- The attack goal: The attacker may force the learner into pulling or avoiding target arms, or worsen the learner's regret, or make the learner identify the wrong best-arm, etc.
- The attack action: The attacker can manipulate the rewards or corrupt the context for contextual bandits, etc.
- Online vs. offline: An online attacker must choose the attack action in real time; An offline attacker poisons a dataset of historical action-reward pairs in batch mode, then the learner learns from the poisoned dataset.

The combination of these attack dimensions presents fertile ground for future research into both bandit-algorithm attacks and the corresponding defense mechanisms.

### Acknowledgments

This work is supported in part by NSF 1837132, 1545481, 1704117, 1623605, 1561512, and the MADLab AF Center of Excellence FA9550-18-1-0166.

## Footnotes

[1]The opposite strategy is to push up the target arm: $\alpha_t = \mathbb{1}\{I_t = K\} \cdot (\mu_K - \max_j \mu_j - \epsilon)$ to make arm $K$ the best arm in post-attack rewards. However, a successful attack means that Alice pulls the target arm $T - o(T)$ times; the attack cost is necessarily linear in $T$, which is inefficient. Simulations that support "drag down" instead of "push up" are presented in Appendix D.

[2]For near-optimal algorithms like UCB [6], one can find the optimal choice of $\epsilon$. See our supplementary material for detail.

[3] One can drop this condition by considering slightly larger $\widetilde{N}(t)$ and smaller $\widetilde{N}_K(t)$. However, we keep the condition as it simplifies $\widetilde{N}(t)$ and $\widetilde{N}_K(t)$. We refer to the proof of Lemma 4 for detail.

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
