[Supplementary Material]

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

)$ only as otherwise $\lambda$ is greater than 1. One can get rid of such a condition by a slightly looser bound. Specifically, using $\lambda = 1$ gives us a bound that holds true for all $T$. We then take the max of the two bounds, which can be simplified as $\sum_{t=1}^{T} X_t < (e-1)\sum_{t=1}^{T}\frac{\epsilon_t}{K} + \sqrt{3\sum_{t=1}^{T}\frac{\epsilon_t}{K}\log\frac{K}{\delta}} + \log\frac{K}{\delta}$ . The condition on $T$ in Theorem 1 can be removed using this bound. However, by keeping the mild assumption on $T$ we keep the exposition simple.

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

## Supplementary Material

## A  Details on the oracle and constant attack

**Logarithmic regret and the suboptimal arm pull counts.**   For simplicity, denote by $i^*$ the *unique* best arm; that is, $i^* = \arg\max_{i=1,\ldots,K} \mu_i$. We show that a logarithmic regret bound implies that the arm pull count of arm $i \neq i^*$ is at most logarithmic in $T$.

**Lemma 7.** *Assume that a bandit algorithm enjoys a regret bound of $O(\log(T))$. Then, $\mathbb{E}N_i(T) = O(\log(T)), \forall i \neq i^*$.*

*Proof.* The logarithmic regret bound implies that for a large enough $T$ there exists $C > 0$ such that $\sum_{i=1}^{K} \mathbb{E}N_i(T)(\mu_{i^*} - \mu_i) \leq C \log T$. Therefore, for any $i \neq i^*$, we have $\mathbb{E}N_i(T)(\mu_{i^*} - \mu_i) \leq C \log T$, which implies that

$$\mathbb{E}N_i(T) \leq \frac{C}{\mu_{i^*} - \mu_i} \log T = O(\log T).$$

$\square$

**Proof of Proposition 1**   By Lemma 7, a logarithmic regret bound implies that the bandit algorithm satisfies $\mathbb{E}N_i(T) = O(\log(T))$. That is, for a large enough $T$, $\mathbb{E}N_i(T) \leq C_i \log(T)$ for some $C_i > 0$. Based on the view that the oracle attack effectively shifts the means $\mu_1, \cdots, \mu_K$, the best arm is now the $K$-th arm. Then, $\mathbb{E}N_K(T) = T - \sum_{i \neq K} \mathbb{E}N_i(T) \geq T - \sum_{i \neq K} C_i \log T = T - o(T)$, which proves the first statement.

For the second statement, we notice that $\mathbb{E}N_i(T) = C_i \log T$ for any $i \neq K$ and that we do not attack the $K$-th arm. Therefore,

$$\mathbb{E}\left[\sum_{t=1}^{T} |\alpha_t|\right] = \sum_{i=1}^{K-1} \mathbb{E}N_i(T) \cdot \Delta_i^\epsilon \leq \sum_{i=1}^{K-1} C_i \Delta_i^\epsilon \log T = O\left(\sum_{i=1}^{K-1} \Delta_i^\epsilon \log T\right).$$

**Proof of Proposition 2**   By Lemma 7, a logarithmic regret bound implies that the bandit algorithm satisfies $\mathbb{E}N_i(T) = O(\log(T))$. Note that the constant attack effectively shifts the means of all the arms by $A$ except for the $K$-th arm. Since $A > \max_i \Delta_i$, the best arm is now the $K$-th arm. Then, $\mathbb{E}N_K(T) = T - \sum_{i=1}^{K-1} \mathbb{E}N_i(T) \geq T - \sum_{i=1}^{K-1} C_i \log T = T - o(T)$, which proves the first statement.

For the second statement, we notice that $\mathbb{E}N_i(T) = C_i \log T$ for any $i \neq K$, and we do not attack the $K$-th arm. Therefore,

$$\mathbb{E}\left[\sum_{t=1}^{T} |\alpha_t|\right] = \sum_{i=1}^{K-1} \mathbb{E}N_i(T) \cdot A \leq A \sum_{i=1}^{K-1} C_i \log T = O(AK \cdot \log T).$$

**The best $\epsilon$ for Alice's oracle attack**   Consider the case where Bob employs a near-optimal bandit algorithm such as UCB [6], which enjoys $\mathbb{E}N_i(T) = \Theta(1 + \sigma^2 \Delta_i^{-2} \log T)$. When the time horizon $T$ is known ahead of time, one can compute the best $\epsilon$ ahead of time. Hereafter, we omit unimportant constant factors for simplicity. Since Alice employs the oracle attack, Bob pulls each arm $1 + \sigma^2 \epsilon^{-2} \log(T)$ times for some $C > 0$ in expectation. Assuming that the target arm is $K$, the attack cost is

$$\sum_{i=1}^{K-1} \Delta_i^\epsilon \cdot (1 + \sigma^2 \epsilon^{-2} \log(T)) = \sum_{i=1}^{K-1} \Delta_i + (K-1) \cdot \epsilon + \sum_{i=1}^{K-1} \left(\frac{\Delta_i}{\epsilon^2} + \frac{1}{\epsilon}\right) \sigma^2 \log T$$

To balance the two terms, one can see that $\epsilon$ has to grow with $T$ and the term $\Delta_i/\epsilon^2$ is soon dominated by $1/\epsilon$. Thus, for large enough $T$ the optimal choice of $\epsilon$ is $\sigma\sqrt{\log(T)}$, which leads to the attack cost of $O(\sum_{i=1}^{K-1} \Delta_i + \sigma K \sqrt{\log T})$.

# B    Details on attacking the $\epsilon$-greedy strategy

**Lemma 8.** *For $\delta \leq 1/2$, the $\beta(N)$ defined in (2) is monotonically decreasing in $N$.*

*Proof.* It suffices to show that $f(x) = \frac{2\sigma^2}{x} \log \frac{\pi^2 K x^2}{3\delta}$ is decreasing for $x \geq 1$. Note that $\delta \leq 1/2 \leq \frac{K}{3}(\frac{\pi}{e})^2$, thus for $x \geq 1$ we have

$$
\begin{aligned}
f'(x) &= -\frac{2\sigma^2}{x^2} \log \frac{\pi^2 K x^2}{3\delta} + \frac{2\sigma^2}{x} \frac{3\delta}{\pi^2 K x^2} \frac{2\pi^2 K x}{3\delta} \\
&= \frac{2\sigma^2}{x^2}(2 - \log \frac{\pi^2 K x^2}{3\delta}) \leq \frac{2\sigma^2}{x^2}(2 - \log \frac{\pi^2 K}{3\delta}) \\
&\leq \frac{2\sigma^2}{x^2}(2 - \log e^2) = 0.
\end{aligned}
$$

$\square$

**Proof of Corollary 1**    When $T$ is larger than the following threshold:

$$
\frac{K+1}{K}(\sum_{t=1}^{T} \epsilon_t) + \sqrt{12\log(K/\delta)(\frac{K+1}{K} \sum_{t=1}^{T} \epsilon_t)},
$$

we have $\widetilde{N}_K(T) \geq \widetilde{N}(T)$. Because $\beta(N)$ is decreasing in $N$,

$$
\widetilde{N}(T)\beta(\widetilde{N}(T)) + 3\widetilde{N}(T)\beta(\widetilde{N}_K(T)) \leq 4\widetilde{N}(T)\beta(\widetilde{N}(T)). \tag{9}
$$

Due to the the exploration scheme of the strategy,

$$
\sum_{t=1}^{T} \epsilon_t = cK \sum_{t=1}^{T} 1/t \leq cK(\log(T) + 1).
$$

Thus by the definition of $\tilde{N}(T)$,

$$
\tilde{N}(T) \leq c(\log T + 1) + \sqrt{3\log\left(\frac{K}{\delta}\right)c(\log T + 1)}.
$$

For sufficiently large $T$, there exists a constant $c_2$ depending on $c, K, \delta$ to further upper bound the RHS as follows:

$$
c(\log T + 1) + \sqrt{3\log\left(\frac{K}{\delta}\right)c(\log T + 1)} \leq c_2 \log T := \check{N}(T). \tag{10}
$$

Since $N\beta(N)$ is increasing in $N$, combining (9) and (10) we have for sufficiently large $T$,

$$
\widetilde{N}(T)\beta(\widetilde{N}(T)) + 3\widetilde{N}(T)\beta(\widetilde{N}_K(T)) \leq 4\check{N}(T)\beta(\check{N}(T)).
$$

Plugging this upper bound into Theorem 1,

$$
\begin{aligned}
\sum_{t=1}^{T} \alpha_t &< \left(\sum_{i=1}^{K} \Delta_i\right) \check{N}(T) + 4(K-1)\check{N}(T)\beta(\check{N}(T)) \\
&= c_2\left(\sum_{i=1}^{K} \Delta_i\right)\log T + \sqrt{32c_2}(K-1)\sigma \cdot \sqrt{\log T\left(2\log\log T + \log \frac{\pi^2 K c_2^2}{3\delta}\right)}. \tag{11}
\end{aligned}
$$

**Proof of Lemma 1**    Let $\{X_j\}_{j=1}^{\infty}$ be a sequence of *i.i.d.* $\sigma^2$-sub-Gaussian random variables with mean $\mu$. Let $\hat{\mu}_N^0 = \frac{1}{N} \sum_{j=1}^{N} X_j$. By Hoeffding's inequality

$$
\mathbb{P}(|\hat{\mu}_N^0 - \mu| \geq \eta) \leq 2\exp\left(-\frac{N\eta^2}{2\sigma^2}\right).
$$

Define $\delta_{iN} := \frac{6\delta}{\pi^2 N^2 K}$. Apply union bound over arms $i$ and pull counts $N \in \mathbb{N}$,

$$\mathbb{P}\left(\exists i, N : |\hat{\mu}_{i,N}^0 - \mu_i| \geq \beta(N)\right) \leq \sum_{i=1}^{K} \sum_{N=1}^{\infty} \delta_{iN} = \delta.$$

**Proof of Lemma 2**  We show by induction that at the end of any round $t \geq K$ Algorithm 1 maintains the invariance

$$\hat{\mu}_K(t) > \hat{\mu}_i(t), \quad \forall i < K, \tag{12}$$

which forces the learner to pull arm $K$ if $t + 1$ is an exploitation round.

Base case: By definition the learner pulls arm $K$ first, then all the other arms once. During round $t = 2 \ldots K$ the attack algorithm ensures $\hat{\mu}_i(t) \leq \hat{\mu}_K(t) - 2\beta(1) < \hat{\mu}_K(t)$ for arms $i < K$, trivially satisfying (12).

Induction: Suppose (12) is true for rounds up to $t - 1$. Consider two cases for round $t$:

If round $t$ is an exploration round and $I_t \neq K$ is pulled, then only $\hat{\mu}_{I_t}(t)$ changes; the other arms copy their empirical mean from round $t - 1$. The attack algorithm ensures $\hat{\mu}_K(t) \geq \hat{\mu}_{I_t}(t) + 2\beta(N_K(t)) > \hat{\mu}_{I_t}(t)$. Thus (12) is satisfied at $t$.

Otherwise either $t$ is exploration and $K$ is pulled; or $t$ is exploitation – in which case $K$ is pulled because by inductive assumption (12) is satisfied at the end of $t - 1$. Regardless, this arm $K$ pull is not attacked by Algorithm 1 and its empirical mean is updated by the pre-attack reward. We show this update does not affect the dominance of $\hat{\mu}_K(t)$. Consider any non-target arm $i < K$. Denote the last time $\hat{\mu}_i$ was changed by $t'$. Note $t' < t$ and $N_K(t') < N_K(t)$. At round $t'$, Algorithm 1 ensured that $\hat{\mu}_i(t') \leq \hat{\mu}_K(t') - 2\beta(N_K(t'))$. We have:

$$\begin{aligned}
\hat{\mu}_K(t) &= \hat{\mu}_K^0(t) & \text{(arm } K \text{ never attacked)} \\
&> \mu_K^0 - \beta(N_K(t)) & \text{((6) lower bound)} \\
&> \mu_K^0 - \beta(N_K(t')) & \text{(Lemma 8)} \\
&> \hat{\mu}_K(t') - 2\beta(N_K(t')) & \text{((6) upper bound)} \\
&\geq \hat{\mu}_i(t') & \text{(Algorithm 1)} \\
&= \hat{\mu}_i(t)\,.
\end{aligned}$$

Thus (12) is also satisfied at round $t$.

**Proof of Lemma 3**  Without loss of generality assume in round $t$ arm $i$ is pulled and the attacker needed to attack the reward (i.e. $I_t = i$ and $\alpha_t > 0$). By definition (4),

$$\begin{aligned}
\alpha_t &= \hat{\mu}_i(t-1)N_i(t-1) + r_t^0 - (\hat{\mu}_K(t) - 2\beta(N_K(t)))\,N_i(t) \\
&= \sum_{s \in \tau_i(t-1)} (r_s^0 - \alpha_s) + r_t^0 - (\hat{\mu}_K(t) - 2\beta(N_K(t)))\,N_i(t) \\
&= \sum_{s \in \tau_i(t)} r_s^0 - \sum_{s \in \tau_i(t-1)} \alpha_s - (\hat{\mu}_K(t) - 2\beta(N_K(t)))\,N_i(t).
\end{aligned}$$

Therefore, the cumulative attack on arm $i$ is

$$\begin{aligned}
\sum_{s \in \tau_i(t)} \alpha_s &= \sum_{s \in \tau_i(t)} r_s^0 - (\hat{\mu}_K(t) - 2\beta(N_K(t)))\,N_i(t) \\
&= \left(\hat{\mu}_i^0(t) - \hat{\mu}_K(t) + 2\beta(N_K(t))\right) N_i(t).
\end{aligned}$$

One can think of the term in front of $N_i(t)$ as the amortized attack cost against arm $i$. By Lemma 1,

$$\begin{aligned}
\hat{\mu}_i^0(t) &< \mu_i + \beta(N_i(t)) \\
\hat{\mu}_K(t) = \hat{\mu}_K^0(t) &> \mu_K - \beta(N_K(t))
\end{aligned}$$

Therefore,

$$\begin{aligned}
\sum_{s:I_s=i}^{t} \alpha_s &< (\mu_i - \mu_K + \beta(N_i(t)) + 3\beta(N_K(t)))\,N_i(t) \\
&\leq (\Delta_i + \beta(N_i(t)) + 3\beta(N_K(t)))\,N_i(t).
\end{aligned}$$

The last inequality follows from the gap definition $\Delta_i := [\mu_i - \mu_K]_+$.

**Proof of Lemma 4** Fix a non-target arm $i < K$. Let $X_t$ be the Bernoulli random variable for round $T$ being arm $i$ pulled. Then,

$$
\begin{aligned}
N_i(T) &= \sum_{t=1}^{T} X_t \\
\mathbb{E}[X_t] &= \frac{\epsilon_t}{K} \\
\mathbb{V}[X_t] &= \frac{\epsilon_t}{K}(1 - \frac{\epsilon_t}{K}) < \frac{\epsilon_t}{K}.
\end{aligned}
$$

Since $X_t$'s are independent random variables, we may apply Lemma 9 of Agarwal et al. [2], so that for any $\lambda \in [0,1]$, with probability at least $1 - \delta/K$,

$$
\begin{aligned}
\sum_{t=1}^{T}(X_t - \frac{\epsilon_t}{K}) &\leq (e-2)\lambda \sum_{t=1}^{T} \mathbb{V}[X_t] + \frac{1}{\lambda}\log\frac{K}{\delta} \\
&< (e-2)\lambda \sum_{t=1}^{T} \mathbb{E}[X_t] + \frac{1}{\lambda}\log\frac{K}{\delta}.
\end{aligned}
$$

Choose $\lambda = \sqrt{\frac{\log(K/\delta)}{(e-2)\sum_{t=1}^{T}\mathbb{E}[X_t]}}$, and we get that

$$
\begin{aligned}
\sum_{t=1}^{T} X_t &< \sum_{t=1}^{T} \frac{\epsilon_t}{K} + 2\sqrt{(e-2)\sum_{t=1}^{T}\frac{\epsilon_t}{K}\log\frac{K}{\delta}} \\
&< \sum_{t=1}^{T} \frac{\epsilon_t}{K} + \sqrt{3\sum_{t=1}^{T}\frac{\epsilon_t}{K}\log\frac{K}{\delta}} := \tilde{N}(T).
\end{aligned}
$$

The same reasoning can be applied to all non-target arm $i < K$. [4]

The case with the target arm is similar, with the only change that $\mathbb{E}[X_t] > 1 - \epsilon_t$ and $\mathbb{V}[X_t] < \epsilon_t$, leading to the lower bound:

$$
N_K(T) > T - \sum_{t=1}^{T}\epsilon_t - \sqrt{3\sum_{t=1}^{T}\epsilon_t\log\frac{K}{\delta}} =: \tilde{N}_K(T).
$$

Finally, a union bound is applied to all $K$ arms to complete the proof.

## C  Details on attacking the UCB strategy

**Proof of Lemma 5** Fix some $t \geq 2K$. If $N_i(t) \leq 2$ for all $i < K$, then $N_K(t) \geq 2$, which implies $N_i(t) \leq \min\{N_K(t), 2\}$. Thus, (8) holds trivially and we are done.

Now fix any $i < K$ such that $N_i(t) > 2$. As the desired upper bound is nondecreasing in $t$, we only need to prove the result for $t$ where $I_t = i$. Let $t'$ be the previous time where arm $i$ was pulled. Note that $t'$ satisfies $K < t' < t$ as $N_i(t) > 2$, so the attacker has started attacking at round $t'$. This implies that $N_i(t'-1) + 1 = N_i(t') = N_i(t-1) = N_i(t) - 1$.

On one hand, it is clear that after attack $\alpha_{t'}$ was added at round $t'$, the following holds:

$$
\hat{\mu}_i(t') \leq \hat{\mu}_K(t') - 2\beta(N_K(t')) - \Delta_0. \tag{13}
$$

On the other hand, at round $t$, it must be the case that

$$\hat{\mu}_i(t-1) + 3\sigma\sqrt{\frac{\log t}{N_i(t-1)}} \geq \hat{\mu}_K(t-1) + 3\sigma\sqrt{\frac{\log t}{N_K(t-1)}},$$

which is equivalent to

$$\hat{\mu}_i(t') + 3\sigma\sqrt{\frac{\log t}{N_i(t')}} \geq \hat{\mu}_K(t-1) + 3\sigma\sqrt{\frac{\log t}{N_K(t-1)}}.$$

Therefore,

$$3\sigma\sqrt{\frac{\log t}{N_i(t')}} - 3\sigma\sqrt{\frac{\log t}{N_K(t-1)}} \geq \hat{\mu}_K(t-1) - \hat{\mu}_i(t')$$

$$\geq \hat{\mu}_K(t-1) - \hat{\mu}_K(t') + 2\beta(N_K(t')) + \Delta_0$$

$$\geq \Delta_0,$$

where we have used Eqn. 13 in the second inequality, the condition in event $E$ as well as Lemma 8 in the third. Since $\Delta_0 > 0$, we can see that $N_i(t') < N_K(t-1)$, and thus

$$N_i(t) = N_i(t') + 1 \leq N_K(t-1) = N_K(t). \tag{14}$$

Furthermore, since $3\sigma\sqrt{\frac{\log t}{N_K(t-1)}} > 0$, we have $3\sigma\sqrt{\frac{\log t}{N_i(t')}} > \Delta_0$, which implies

$$N_i(t) = 1 + N_i(t') \leq 1 + \frac{9\sigma^2}{\Delta_0^2}\log t. \tag{15}$$

Combining (14) and (15) gives the desired bound (8).

**Proof of Lemma 6**   Fix any $i < K$. As the desired upper bound is increasing in $t$, we only need to prove the result for $t$ where $I_t = i$ and $\alpha_t > 0$. It follows from (7) that,

$$\frac{1}{N_i(t)}\sum_{s\in\tau_i(t)}\alpha_s = \hat{\mu}_i^0(t) - \hat{\mu}_K(t-1) + 2\beta(N_K(t-1)) + \Delta_0.$$

Since event $E$ holds, we have

$$\frac{1}{N_i(t)}\sum_{s\in\tau_i(t)}\alpha_s \leq \Delta_i + \Delta_0 + \beta(N_i(t)) + 3\beta(N_K(t-1)).$$

The proof is completed by observing $N_K(t-1) = N_K(t)$, $N_i(t) \leq N_K(t)$ (Lemma 5) and Lemma 8.

## D   Simulations on Heuristic Constant Attack

We run simulations on $\epsilon$-greedy and UCB to illustrate the heuristic constant attack algorithm. The bandit has two arms, where the reward distributions are $\mathcal{N}(1, 0.1^2)$ and $\mathcal{N}(0, 0.1^2)$ respectively, thus $\max_i \Delta_i = \mu_1 - \mu_2 = 1$. Alice's target arm is arm 2. In our experiment, Alice tried two different constants for attack: $A = 1.2$ and $A = 0.8$, one being greater and the other being smaller than $\max_i \Delta_i$. We run the attack for $T = 10^4$ rounds. Fig. 3 and Fig. 4 show Alice's cumulative attack cost and Bob's number of target arm pulls $N_K(t)$ for $\epsilon$-greedy and UCB. Note that if $A > \max_i \Delta_i$, then $N_K(t) \approx t$, which verifies that Alice succeeds with the heuristic constant attack. At the same time, pushing up the target arm would incur linear cost; while dragging down the non-target arm achieves logarithmic cost. In summary, Alice should use an $A$ value larger than $\Delta$, and should drag down the expected reward of the non-target arm by amount $A$.

(a) push up the target arm: $\alpha_t = \mathbb{1}\{I_t = 2\} \cdot (-A)$     (b) drag down the non-target arm: $\alpha_t = \mathbb{1}\{I_t \neq 2\} \cdot A$

Figure 3: Constant attack on $\epsilon$-greedy

(a) push up the target arm                (b) drag down the non-target arm

Figure 4: Constant attack on UCB1