[Reviews · NeurIPS 2018]

Reviewer 1



This paper studies the reward manipulation attacks on stochastic bandit settings. Here, the main purpose is to guarantee a successful attack. Theorem 1 and Theorem 2 claim that an attack is guaranteed for e-greedy and UCB algorithms respectively. In my opinion, the paper could provide more background for the readers who are not familiar with the attacks on bandits. What should the attacker do: e.g force the learner into pulling or avoiding target arms, or worsen the learner’s regret ...? Besides, they could discuss what could be done to defend against the attacks? Secondly, the authors can mention before the Theorem1 that it is a typical high-probability statement and the 'failure probability' parameter _x000e_is usually set close to 0 and the theorems involving high-probability statements are abundant in the literature. Overall, I like the idea in this paper and I'm sure about the correctness of the theorems. If the authors can clarify some points that I've mentioned, this could be a strong paper. It also opens a new research area in the literature.

Reviewer 2



This paper studies adversarial attacks on stochastic bandits. Though adversarial learning has been studied in deep learning, this is the first study of attacks on stochastic bandit algorithms to the best of my knowledge. They propose the attacks against two popular bandit algorithms, epsilon-greedy and UCB, without knowing the ground truth. Their attacks can make the bandit algorithm to treat a certain suboptimal action as the optimal action with only O(log T) cost. The paper is well written and the idea is quite novel. This work may be a seminal work that can inspire a line of research in attacks on bandits and robust bandit algorithms. However, they do not propose any defense for these attacks. It would be more interesting to design robust algorithms. ====== after rebuttal ====== I read the rebuttal.

Reviewer 3



This paper presents adversarial attacks against multi-arm bandit algorithms e-greedy and UCB. This paper is very well written; the problem is well-motivated; the approach is sound and well explained through the guidance from an oracle approach in an less realistic setting to the real setting. I enjoy reading this paper. I list some of the aspects that can be improved below and hope they can help the authors to improve the paper. But, again, I think the paper is publishable even in the current version. First, it would be better if the threat model can be further justified, i.e., why the attacker can manipulate the rewards arbitrarily? It would be good if the authors can map the attacker to real settings. Second, I don't find how the attacks can be mitigated if Bob is aware of the existence of the attacker. This would be an interesting discussion to see to what can be an equilibrium between the attacker and the defender. Third, which is related to the above aspects, the authors only consider two bandit algorithms, leaving a large space of other bandit algorithms not discussed. Thus, it is unclear whether Bob can simply choose another algorithm to defend against the attacks? Would there be any universal results that can be said about the attacker without relying on the concrete bandit algorithm chosen by the defender? The authors admit that the work is still in progress, and many more bandit algorithms will be discussed; I hope, in the future versions, there could be some general conclusions presented in par of these concrete bandit algorithms, which might provide more tight results.